# Development of a Ferritin-Based Nanoparticle Vaccine against Classical Swine Fever

**DOI:** 10.3390/vaccines12080948

**Published:** 2024-08-22

**Authors:** Yiwan Song, Zhongmao Yuan, Junzhi Ji, Yang Ruan, Xiaowen Li, Lianxiang Wang, Weijun Zeng, Keke Wu, Wenshuo Hu, Lin Yi, Hongxing Ding, Mingqiu Zhao, Shuangqi Fan, Zhaoyao Li, Jinding Chen

**Affiliations:** 1College of Veterinary Medicine, South China Agricultural University, No. 483 Wushan Road, Tianhe District, Guangzhou 510642, China; songyiwan1111@163.com (Y.S.); zhongmao.yuan@stu.scau.edu.cn (Z.Y.); jijunzhi2022@163.com (J.J.); ruanyang96@163.com (Y.R.); 18306616234@163.com (X.L.); zwj1102727352@stu.scau.edu.cn (W.Z.); 13660662837@163.com (K.W.); huwenshuo1219@163.com (W.H.); yilin@scau.edu.cn (L.Y.); dinghx@scau.edu.cn (H.D.); zmingqiu@scau.edu.cn (M.Z.); shqfan@scau.edu.cn (S.F.); 2Key Laboratory of Zoonotic Disease Prevention and Control of Guangdong, South China Agricultural University, Guangzhou 510642, China; 3Wen’s Group Academy, Wen’s Foodstuffs Group Co., Ltd., Xinxing 527400, China; wanglianxiang@wens.com.cn

**Keywords:** ferritin nanoparticles, classical swine fever virus, E2 protein, B-cell antigenic epitope, immune responses

## Abstract

The occurrence of classical swine fever (CSF) poses a significant threat to the global swine industry. Developing an effective and safe vaccine is crucial for preventing and controlling CSF. Here, we constructed self-assembled ferritin nanoparticles fused with the classical swine fever virus (CSFV) E2 protein and a derived B cell epitope (Fe-E2B) using a baculovirus expression system (BVES), demonstrating enhanced immunogenicity. Furthermore, we provide a detailed evaluation of the immunological efficacy of the FeE2B in rabbits. The results showed that robust and sustained antibody responses were detected in rabbits immunized with the Fe-E2B nanoparticle vaccine, comparable to those elicited by commercially available vaccines. Additionally, we demonstrated that the vaccine effectively activated crucial immune factors IFN-γ and IL-4 in vivo, increasing their levels by 1.41-fold and 1.39-fold, respectively. Immunization with Fe-E2B enabled rabbits to avoid viremia and stereotypic fever after CSFV challenge. In conclusion, this study highlights the potential of ferritin nanoparticles as antigen-presenting carriers to induce robust immune responses, proposing a candidate vaccine strategy for the prevention and control of CSF.

## 1. Introduction

Classical swine fever (CSF) is a highly contagious viral disease caused by the classical swine fever virus (CSFV), characterized by high fever, multi-system hemorrhagic lesions, and immunosuppression [1]. It is classified as a notifiable disease by the World Animal Health Organization [2]. CSFV is a single-stranded positive-sense RNA-enveloped virus belonging to the genus *Pestivirus* of the family *Flaviviridae* [3]. The virus remains prevalent in several countries, including regions in Asia, Eastern Europe, and South America, causing substantial economic losses to the global pig industry [4]. Therefore, the research and development of a CSF vaccine is of great socio-economic significance. Currently, the predominant vaccines against CSFV on the market are live attenuated vaccines [5] and subunit vaccines [6]. However, the primary drawback of live attenuated vaccines is the absence of serological markers, which hinders the differentiation between infected and vaccinated animals (DIVA) [7,8]. Subunit vaccines, on the other hand, require multiple injections or high doses yet still exhibit comparatively weaker immune responses [9].

CSFV has a single open reading frame (ORF) encoding a precursor polyprotein, which is further translated and processed into four structural proteins (C, E^rns^, E1, and E2) and eight non-structural proteins (N^pro^, p7, NS2, NS3, NS4A, NS4B, NS5A, and NS5B) [10]. Among the surface proteins of CSFV, the structural glycoprotein E2 exhibits the strongest immunogenicity, inducing high levels of neutralizing antibodies that can protect pigs against lethal challenge [11,12]. However, subunit vaccines prepared with the E2 glycoproteins often require multiple injections and high doses to achieve the same immunization effect as C-strain, which may be accompanied by cost issues [13,14]. Therefore, improving the immunogenicity of antigens is a core issue in developing effective and practical subunit vaccines against CSFV. Epitopes, as the minimal immune regions on viral proteins, can precisely induce immune responses in vivo [15,16]. The E2 glycoprotein contains numerous conserved epitopes, including the B-cell antigenic epitope TAVSPTTLR (amino acids 829–837), which elicit anti-CSFV neutralizing antibodies in pigs [17,18,19]. To enhance immunogenicity, a vaccine was prepared in this study using a fusion expression of the CSFV E2 protein and the TAVSPTTLR epitope as an immunogen.

The application of nanoparticles represents a significant breakthrough in the medical field [20,21,22,23,24]. Nanoparticles have been used for antigen presentation and immune stimulation in vaccine development [25,26], including for COVID-19 vaccines [27,28,29] and the Zika virus vaccine [30,31,32]. Various nanoparticles, including ferritin, have recently been explored and applied in vaccine delivery systems [33,34]. Ferritin is an iron storage protein widely present in most organisms that self-assembles into nearly spherical nanoparticles composed of 24 subunits with an octahedral appearance and a hollow interior [35,36]. When used as an antigen delivery platform, ferritin enhances the efficiency of antigen uptake by antigen-presenting cells (APCs) [37], leading to better stimulation of B-cell maturation into plasma cells secreting high-affinity IgG, with the assistance of T follicular helper (TFH) cells [38,39]. Furthermore, ferritin possesses structural stability, and the insertion of appropriate exogenous proteins does not affect its self-assembly [21]. The baculovirus expression vector system (BVES) is a high-yield eukaryotic gene expression system characterized by its large capacity for exogenous protein expression, efficient post-translational modifications, high bioactivity of expressed proteins, and a high safety profile [40]. It can be a candidate expression system for ferritin.

In this study, self-assembling ferritin nanoparticles fused with exogenous proteins were constructed by inserting CSFV E2 glycoprotein and its TAVSPTTLR epitope onto the surface of ferritin nanoparticles through genetic modification. It is well known that the C-strain was obtained by hundreds of passages of the highly virulent CSFV strain in rabbits [41,42]. After inoculating the C-strain in the ear vein, rabbits exhibited typical clinical symptoms of fever lasting 18–24 h. Studies have shown that the CSFV can extensively replicate in the spleen of experimental rabbits [43,44,45], with the E2 protein and E^rns^ protein being critical factors for maintaining viral replication in rabbits [46]. This indicates that rabbits are susceptible to CSFV. Therefore, considering cost and safety factors, rabbits were selected in this study to evaluate the immune efficacy of the recombinant ferritin nanoparticle vaccine against CSFV as a substitute for pigs. The results of this study show that the recombinant ferritin nanoparticle vaccine efficiently induces cellular and humoral immunity. This suggests that this approach could be a viable measure for preventing and controlling CSF in pigs.

## 2. Materials and Methods

### 2.1. Cells and Viruses

Spodoptera frugiperda (Sf9) cells and High Five (Hf5) cells were obtained from the China Typical Culture Preservation Center. Porcine kidney (PK-15) cells, CSFV C-strain, pFastBac Dual (pFBD), and DH10Bac strains were preserved by the Department of Microbiology and Immunology, College of Veterinary Medicine, South China Agricultural University. PK-15 cells were cultured in high-glucose Dulbecco’s Modified Eagle Medium (Gibco, Thermo Fisher Scientific, Carlsbad, CA, USA) supplemented with 10% (*v*/*v*) fetal bovine serum (ExCell Bio-Tech Co., Suzhou, China). Sf9 cells were cultured within SIM SF medium (Sino Biological, Beijing, China) in an incubator at 27 °C (Thermo Fisher Science, Carlsbad, CA, USA), and Hf5 cells were cultured within SIM HF medium (Sino Biological, Beijing, China) at 27 °C, 140 rpm/min in a shaker (Thermo Fisher Science, Carlsbad, CA, USA).

### 2.2. Construction of Recombinant Transfer Plasmids

The CSFV E2 protein nucleotide sequence (GenBank accession number: NC001847.1) and the *Helicobacter pylori* ferritin amino acid sequence (GenBank accession number: NP_223316) were obtained from the National Center for Biotechnology Information (NCBI). A novel nucleotide recombinant gene sequence (E2B) was constructed by inserting a nucleotide sequence of linear B-cell epitope TAVSPTTLR to the N-terminus of the E2 protein nucleotide sequence. The recombinant gene Fe-E2B was subsequently constructed by linking the E2B sequence to the Helicobacter pylori ferritin gene sequence using a flexible linker peptide (GGGGS). These recombinant gene sequences were then inserted into the pFBD vector by *Bam*HI and *Hin*dIII restriction sites, with a base sequence encoding six histidine (6 × His) added to the terminal ends, resulting in the creation of the pFBD-E2, pFBD-E2B, and pFBD-Fe-E2B transfer plasmids. These three recombinant plasmids were synthesized by Tsingke (Tsingke Biotechnology Co., Ltd., Beijing, China) and optimized for insect cell codon preferences.

### 2.3. Expression and Identification of Recombinant Baculovirus

The transfer plasmids were transformed into DH10Multibac cells, followed by blue-white screening and purification steps to harvest recombinant baculovirus plasmids. These plasmids were then transfected into Sf9 cells at a density of 1 × 10^6^ cells/mL using Cellfectin II Reagent (Thermo Fisher Scientific Inc., Waltham, MA, USA). After five days, the supernatant was collected and stored at −80 °C as a viral seed stock F1-generation, and the cells were harvested for identification via Western blotting. The harvested viral fluids were re-inoculated with Sf9 cells. After three days of expression, they were characterized by immunofluorescence assay (IFA) using His-tag antibodies (Sino Biological, Beijing, China) as primary antibodies.

### 2.4. Expression and Identification of Recombinant Protein

Recombinant baculoviruses were inoculated into Hf5 cells at 2 × 10^6^ cells/mL density. After five days, cell suspensions were collected and validated using Western blotting. Recombinant proteins were separated on a 12.5% SDS-PAGE gel, transferred to a polyvinylidene fluoride (PVDF) membrane (Millipore, Burlington, MA, USA), and blocked with 5% (*w*/*v*) skim milk in Phosphate-Buffered Saline (PBS) containing 0.01% Tween (PBST) for 1 h at 37 °C. PVDF membranes were then incubated overnight with His-tag antibodies (Sino Biological, Beijing, China) and murine anti-E2 monoclonal antibodies (WH303 Monoclonal Antibody to Pestiviruses, 1:500, APHA Scientific, Surrey, UK) in primary antibody dilution buffer (1:1000, Beyotime, Guangzhou, China). Following three washes with PBST, membranes were incubated with HRP-conjugated goat anti-mouse antibodies (1:1000, Abbkine, Wuhan, China) diluted in PBST for 1 h at 37 °C. The chemiluminescent signal was developed using Super Signal West Pico/Femto chemiluminescent substrate (Thermo Fisher Scientific, Waltham, MA, USA) and visualized with a Tanon 5200 Luminescence Imaging System (Tanon, Shanghai, China).

### 2.5. Optimization of Protein Expression Conditions and Protein Purification

Recombinant baculoviruses were propagated on Sf9 cells for two generations and harvested as F3-generation viral fluids. The F3-generation recombinant baculovirus was inoculated into Hf5 cells at various multiplicities of infection (MOI) (MOI = 1, MOI = 5, and MOI = 10). Cell suspensions were collected at different time points (48 h, 96 h, 72 h, and 120 h) for analysis using Western blotting. Specific bands detected with Western blotting were quantified using ImageJ 2 (National Institutes of Health; Bethesda, MD, USA). Under optimal protein expression conditions, large volumes of Hf5 cell suspensions were obtained, and cells were harvested by centrifugation at 1000× *g* for 10 min. Hf5 cells were resuspended in 20 mL of PBS and subjected to sonication on ice, with 5 s pulses and 9 s pauses for 40 min at 200 W. The lysate was then centrifuged at 15,000× *g* for 20 min at 4 °C. The supernatant was collected and filtered through a 0.45 μm membrane for further use. A series of sucrose gradients (70%, 40%, and 20%) were sequentially layered in ultracentrifuge tubes (Beckman Coulter, Brea, USA). The filtered supernatant was then added to the tubes and centrifuged at 40,000× *g* for 8 h at 4 °C.

### 2.6. Characterization by Transmission Electron Microscopy

Diluted purified recombinant protein samples (10 μL) were applied to carbon-coated grids. After 1 min, the excess sample was blotted with filter paper. The recombinant protein samples were stained with 1% uranyl acetate for 1 min and then washed with deionized water to remove excess impurities. The samples were examined using transmission electron microscopy (TEM) (FEI Company, Hillsboro, OR, USA).

### 2.7. Animals and Animal Welfare

Female New Zealand rabbits, 14 weeks old, were purchased from Southern Medical University. All rabbit care practices and research were approved by the Ethics Committee and the Laboratory Animal Care and Use Committee of South China Agricultural University. Pentobarbital sodium is a medium-acting barbiturate sedative-hypnotic drug that acts rapidly upon intravenous administration. Rabbits show no signs of pain during administration, making it an effective anesthetic. At the end of the experiments, the rabbits were anesthetized with an auricular vein injection of 100 mg/kg pentobarbital sodium (for rabbits weighing >2 kg) and monitored for corneal reflexes. Blood and tissue samples were then harvested for subsequent analyses.

### 2.8. Rabbits’ Immunization and Challenge

New Zealand rabbits were divided into seven groups (N = 5): PBS, Fe, E2, E2B, Fe-E2, Fe-E2B, and C. The Fe, E2, E2B, Fe-E2, and Fe-E2B. The purified protein contained in the vaccine formulation was set at 40 µg according to the relevant literature [47]. Montanide ISA-201 VG adjuvant (Seppic, Air Liquide, France), W/O/W adjuvant, is an excellent vaccine adjuvant that induces a faster immune response and produces high levels of neutralizing antibodies [48,49]. The vaccine formulation was prepared by mixing ISA 201VG adjuvant and recombinant protein in a 1:1 ratio and stirring at 600 r/min for 1 h using a magnetic stirrer. The control group was injected with 1 mL of autoclaved PBS. The C-strain vaccine was immunized with 1 dose of C-strain attenuated vaccine (commercial vaccine) (Guangdong Winsum Bio-pharmaceutical Co., Ltd., Guangzhou, China) per rabbit via intramuscular injection (IM). Booster immunizations were carried out with the same dose at 21 days post-first immunization (dpi). At 35 dpi, 1 mL of 100 RID_50_ (50% rabbit infectious dose) of C-strain was intravenously (IV) injected into the rabbits in the experiment. The immunization protocols for the animals are shown in Table 1.

### 2.9. Detection of Specific Antibody Titers by ELISA

Whole blood was collected from rabbits at 7, 14, 21, 35, 42, and 49 dpi. The collected blood samples were incubated at 37 °C for 1 h, then centrifuged at 2000× *g* for 5 min at 4 °C to harvest the sera, which were subsequently labeled. The sera were analyzed for specific antibody responses using a CSFV antibody ELISA kit (Jiangsu Enzyme Immunity Industry Co., Ltd.; Yancheng, China) according to the manufacturer’s instructions. Briefly, 100 μL of diluted serum (1:40) was added to each well and incubated at 37 °C for 30 min. Then, 300 μL of PBS was added to each well, and the plates were washed using an automatic plate washer for 1 min, repeated three times. Enzyme-labeled goat anti-rabbit immunoglobulin was added to the wells, followed by incubation for 30 min at 37 °C. The HRP signal was detected using a tetramethylbenzidine (TMB) substrate. The reaction was stopped by adding 50 μL of 2 M sulfuric acid, and OD values were measured at 450 nm using an ultraviolet spectrophotometer (Eppendorf, Hamburg, Germany).

### 2.10. Detection of Neutralizing Antibody Titers

Sera were separated at 35 and 49 dpi and then inactivated at 56 °C for 30 min in a water bath. Serial 2-fold diluted sera and 200 TCID_50_/100 μL of the CSFV C-strain were mixed in equal proportions in 96-well microtiter plates with three replicates. Subsequently, 100 μL of well-grown PK-15 cells at a concentration of 3 × 10^5^ cells/mL were added and incubated for 48 h in a 37 °C, 5% CO_2_ incubator. Neutralizing antibodies were detected using an indirect IFA with an anti-E2 monoclonal antibody (1:1500, APHA, UK) and FITC-labeled goat anti-rabbit IgG (1:26,000, Beyotime Biotechnology, Shanghai, China). The results were observed under a fluorescence microscope (Nikon Corporation, Tokyo, Japan).

### 2.11. Lymphocyte Proliferation Assay

Peripheral blood mononuclear cells (PBMCs) were isolated from whole blood collected at 35 and 49 dpi using a PBMC isolation kit (Jiangsu Enzyme Immunity Industry Co., Ltd., Yancheng, China). The isolated PBMCs were cultured in RPMI 1640 complete medium (Gibco, Carlsbad, CA, USA) with the addition of 10% fetal bovine serum (ExCell Bio-Tech Co., Shanghai, China) and incubated in 96-well microtiter plates at 100 μL per well. Each set of 96-well microtiter plates was supplemented with 100 μL of 10^5^TCID_50_ CSFV as the stimulus antigen, 100 μL of Con A (Solarbio Life Science, Beijing, China) at a final concentration of 10 μg/mL as the positive control, and 100 μL of 1640 complete medium as the negative control. The plates were then incubated for 24 h in a 37 °C, 5% CO_2_ incubator. Subsequently, 10 μL of CCK-8 solution (Cell Counting Kit-8, Dojindo, Japan) was added to each well, and the plates were incubated for an additional 4 h under the same conditions. Finally, the absorbance at 450 nm was measured using an ultraviolet spectrophotometer (Eppendorf, Germany). Lymphocyte proliferation stimulation index (SI) = average OD 450 nm values of CSFV-stimulated wells/average OD 450 nm values of complete medium-treated wells.

### 2.12. Serum Cytokine Levels Were Detected by ELISA

Sera were taken from each group of immunized rabbits at 35 and 49 dpi. The Rabbit IFN-γ/IL-4 ELISA kit (Jiangsu Enzyme Immunity Biotech Co., Ltd., Nanjing, China) was used to detect cytokines IFN-γ and IL-4 levels.

### 2.13. Viral Load Detection by qPCR

All rabbits were euthanized at 49 dpi, and lymph nodes, spleen, and peripheral blood were collected. RNA was extracted using the RNA isolation kit (Omega Bio-Tek, WikiPredia, Norcross, GA, USA) and reverse transcribed into cDNA with Super Script IV (Thermo Fisher Scientific, Waltham, MA, USA). Viral loads in the lymph nodes, spleen, and peripheral blood were quantified using the CSFV TaqMan quantitative polymerase chain reaction (qPCR) assay developed in our laboratory.

### 2.14. Data Analysis

All data in this study were expressed as mean ± standard deviation (SD), and a one-way or two-way ANOVA was performed using Graph Pad Prism 8.0 software (Graph Pad Software, La Jolla, CA, USA). Subsequently, multiple comparisons were used to compare between groups. A *p* value of <0.05 was considered to indicate a significant difference. Ns, no significant differences were observed (ns, *p* > 0.05; *, *p* < 0.05; **, *p* < 0.01; ***, *p* < 0.001; ****, *p* < 0.0001).

## 3. Results

### 3.1. Construction and Identification of the Recombinant Ferritin Nanoparticles

To investigate the expression and antigenicity of recombinant proteins, pFBD-E2B, pFBD-E2B, pFBD-Fe-E2B, and pFBD-Fe-E2 transfer plasmids were constructed as shown in Figure 1A. The successful rescue of recombinant baculovirus Ac-E2B, Ac-Fe-E2, and Ac-Fe-E2B were confirmed by IFA, as shown in Figure 1B,C. Sf9 cells containing recombinant baculovirus emit intense green fluorescence under a fluorescence microscope (Nikon Corporation, Tokyo, Japan) following treatment with anti-His tag antibodies (Figure 1B) and anti-E2 antibody (Figure 1C). Furthermore, Western blotting results demonstrated the successful expression of the Ac-E2B protein (48 kDa), Ac-Fe-E2 protein (64 kDa), and Ac-Fe-E2B protein (67 kDa) (Figure 1B,C). The combination of Western blotting and IFA experiments confirmed the antigenicity of the recombinant proteins. The Western blotting results showed specific protein bands at molecular weights of 48 kDa, 64 kDa, and 67 kDa, corresponding to the expected sizes of the Ac-E2B, Ac-Fe-E2, and Ac-Fe-E2B proteins, respectively, indicating successful construction of the recombinant proteins (Figure 1B,C). In the Western blotting results for Ac-Fe-E2 and Ac-Fe-E2B, two protein bands with molecular weights of 44 kDa and 48 kDa that were lower than the target bands were observed. These bands likely resulted from the breakage of the GGGGS linker during the expression of recombinant proteins. However, this did not interfere with the formation of recombinant ferritin nanoparticles or subsequent experiments.

### 3.2. Expression and Purification of the Recombinant Protein

To achieve high yields of recombinant protein, this study optimized protein expression conditions by adjusting the recombinant baculovirus’s MOI and the protein expression duration. Western blotting and grayscale analysis results indicated that the optimal protein expression conditions were an MOI of 1 and 96 h for E2B (Figure 2A), an MOI of 1 and 120 h for Fe-E2 (Figure 2B), and an MOI of 1 and 96 h for Fe-E2B (Figure 2C). Hf5 cell suspensions, fragmented by ultrasound, were added onto ultracentrifuge tubes containing 20%, 40%, and 70% sucrose density gradients and centrifuged at 40,000× *g* for 8 h. Due to the difference in sucrose concentration and protein properties, this process resulted in the formation of three distinct protein bands (upper, middle, and lower protein bands) in the gradient layers (Figure 2D), which were subsequently collected and analyzed using SDS-PAGE. The results (Figure 2E–G) demonstrated that the bands in the lanes containing the target proteins were relatively distinct after sucrose gradient centrifugation, indicating suitable purification. Additionally, this method removed excess fragmented protein from the recombinant proteins Fe-E2 and Fe-E2B.

### 3.3. Ferritin Nanoparticles Were Identified by TEM

The purified recombinant proteins were stained with 1% uranyl acetate and were observed using TEM. Visible nanoparticles were observed in all recombinant proteins except for recombinant protein E2B (Figure 3). Furthermore, the recombinant proteins Fe-E2 (Figure 3C) and Fe-E2B (Figure 3D) displayed significant bulges, indicating the insertion of exogenous proteins on the surface of the ferritin. The TEM results demonstrated the successful construction of ferritin nanoparticles fused with the CSFV E2 protein and B-cell antigenic epitopes via the BVES

### 3.4. Detection of Antibody Titers in the Immunized Rabbits

The ELISA kit containing the CSFV E2 antigen detected specific antibodies produced by each vaccine (Figure 4A). There was no difference in the specific antibody levels between the PBS group and the Fe group, indicating that ferritin does not stimulate a specific B-cell immune response. At 28 dpi, the E2B, Fe-E2, and Fe-E2B groups all induced high levels of specific antibodies, which peaked at 49 dpi, showing no significant difference compared to the C-strain vaccine (Figure 4A, *p* > 0.05). The ferritin-based vaccine groups induced higher specific antibody titers than non-ferritin. Specifically, the Fe-E2 group exhibited higher specific antibody titers than the E2 group, and the Fe-E2B group showed higher specific titers than the E2B group. Although the differences between the Fe-E2 and E2 groups, as well as between the Fe-E2B and E2B groups, were not statistically significant, the mean values of Fe-E2 and Fe-E2B were higher than those of the E2 and E2B groups, respectively, as shown in Figure 4. Furthermore, the Fe-E2B group has higher antibody titers than the E2B group at 42 dpi (Figure 4A, *p* < 0.05). Notably, among all experimental groups (except for the C-strain vaccine), the Fe-E2B group induced the highest levels of specific antibodies (Figure 4A). At 49 dpi, the level of specific antibodies in the Fe-E2B vaccine group was higher than those in the C-strain vaccine.

Sera collected at 35 and 49 dpi were used to determine neutralizing antibody titers in porcine serum using the fixed virus-diluted sera method (Figure 4B). There were no statistically significant differences in neutralizing antibody titers among the vaccine groups. However, the Fe2B group elicited the highest neutralizing antibody titers, which were slightly higher than the C group and significantly higher than the PBS group (*p* < 0.0001).

### 3.5. Detection of the Proliferation of Lymphocytes and the Levels of Cytokines in Rabbits

To explore cellular immune responses, we isolated PBMCs from experimental rabbits and assessed lymphocyte proliferation and cytokine levels. The ferritin-based vaccine group induced a higher lymphocyte proliferation compared to the non-ferritin group. (Figure 5A, 49 dpi, Fe-E2B/E2B, *p* < 0.01/*p* < 0.05). Among all vaccine groups, except for the C-strain vaccine, the Fe-E2B group exhibited the highest lymphocyte proliferation (Figure 5A).

To further investigate the cellular immunity mechanisms induced by recombinant ferritin nanoparticle vaccines, we measured the levels of IL-4 and IFN-γ in the sera. The Fe-E2B group demonstrated the highest IFN-γ levels among all vaccine groups (Figure 5B), significantly higher than the PBS group (*p* < 0.0001) and the E2B group (35 dpi, *p* < 0.01; 49 dpi, *p* < 0.05) and slightly higher than the C-strain vaccine (*p* < 0.05, 35 dpi). Additionally, the Fe-E2B and Fe-E2 groups exhibited significantly higher IL-4 levels compared to the PBS group (Figure 5C, *p* < 0.001), with no significant difference compared to the C-strain vaccine (Figure 5C).

### 3.6. Rectal Temperature Changes in Rabbits after Challenge

To evaluate the recombinant ferritin nanoparticle vaccine’s ability to prevent fever in challenged rabbits, we administered 100 RID_50_ of the C-strain into the ear vein at 35 dpi and recorded their body temperatures. Temperature changes were measured continuously for 120 h at 12-h intervals, and temporal temperature change curves were plotted. The results (Figure 6) show that rabbits in the Fe and PBS groups developed a stereotyped fever from 12 to 108 h post-challenge (hpi). In contrast, the body temperatures of rabbits in all other immunized groups remained within the normal range (38.5–40 °C) post-challenge, except for one rabbit in each of the E2, Fe-E2, and Fe-E2B groups, which exhibited a transient increase.

### 3.7. Viral Load Was Measured by qPCR

The vaccine’s efficacy in clearing the viral infection was verified by measuring the viral load in experimental rabbits. The rabbits were euthanized at 49 dpi, and their lymph nodes, spleen, and peripheral blood were collected. Tissues from three randomly selected rabbits were used to extract viral RNA, which was then reverse-transcribed. The viral loads in these tissues were quantified using qPCR. The qPCR results (Table 2) showed detectable CSFV nucleic acid in the lymph nodes, spleen, and peripheral blood in both the PBS and Fe groups. These data indicate that the vaccine constructed for our experiment effectively protected the rabbits against viremia following the C-strain challenge.

## 4. Discussion

CSFV has been eradicated in some European countries by immunizing pigs with live attenuated vaccines and culling CSFV-infected pigs. However, new outbreaks occasionally occur in previously disease-free areas, indicating repeated reintroduction of the virus [50,51]. The conventional live attenuated C-strain vaccines widely used on the market today lack the capability for DIVA [52]. In developing countries, culling infected pigs is not an economically viable measure. The E2 protein, the major antigenic protein of CSFV, has been designed as a DIVA vaccine target. However, the E2 protein has not proven as effective at immunization as the C-strain live attenuated vaccine [53,54,55]. Therefore, it is necessary to identify a novel vaccine for effectively controlling and eradicating CSF. In this study, we designed a recombinant ferritin nanoparticle vaccine using E2 protein and B cell antigen epitopes as antigens and ferritin as a carrier. The vaccine induced a robust immune response in the rabbit model, demonstrating its potential for development and application.

The ability to produce high levels of high-affinity immunoglobulin G (IgG) produced by plasma cells is one of the most important indicators for assessing the immunologic efficacy of a vaccine [56]. In this study, all rabbits in the experimental groups, except those in the PBS and Fe groups, exhibited an immune response in vivo. Effective antibody titers were detected (Figure 4). Rabbits in the PBS and Fe groups exhibited typical fever 12 h after the challenge. In contrast, the body temperatures of rabbits in the other experimental groups remained essentially normal. In the E2, Fe-E2, and Fe-E2B groups, some rabbits experienced temporary increases in body temperature, which later returned to normal (Figure 6). This temporary increase in body temperature might be attributed to non-specific stress factors. No CSFV was detected in rabbits’ lymph nodes, peripheral blood, or spleen in the vaccine groups, indicating that the vaccine prepared for this experiment can eliminate viremia and typical fever following challenge in rabbits.

Importantly, ferritin can be efficiently recognized by antigen-presenting cells and presented as a 24-subunit polymer [57]. In this experiment, ferritin nanoparticles were constructed using a BVES. The CSFV E2 protein and B-cell antigenic epitopes were presented on the nanoparticles to generate higher humoral immunity titers. The ferritin-based vaccine group induced higher specific antibody titers in the early immunization stage (Figure 4A). The neutralizing antibody titers in the Fe-E2 group were 1.2 times higher than those in the E2 group (Figure 4B, 14 dpi).

T cells can differentiate into effector and memory T cells under antigenic stimulation. Effector T cells further differentiate into helper T cell subsets with distinct functions, namely helper T cell types 1 and 2 (Th1 and Th2, respectively). Th1 cells release cytokines, particularly IFN-γ, which promote lymphocyte proliferation and induce cell-mediated immune responses. In contrast, Th2 cells release cytokines such as IL-4, which encourage the proliferation and differentiation of B cells to produce antibodies [58]. Interestingly, during the early stages of immunization, IFN-γ acts before humoral immunity following immunization with the C-strain [59]. This suggests that the innate immune factor IFN-γ plays a crucial role in responding to CSFV infection. In our experiments, the vaccine with ferritin nanoparticles as carriers induced cellular immune responses in rabbits via Th1 and Th2 cells (Figure 5B), showing a greater preference for the Th1 type (Figure 5B). This indicates that the recombinant CSFV ferritin nanoparticle vaccines constructed in this experiment effectively activate innate immune factors in vivo.

Epitopes are shorter amino acid sequences in protein antigens more easily recognized by MHC molecules. Compared with the whole protein antigen, they could directly induce effective immunity [60]. The N-terminal presence of the E2 antigen comprises four separate antigenic regions—A, B, C, and D—which contain linear neutralization epitopes [61]. Peng et al. used the monoclonal antibody WH303 to verify that TAVSPTTLR (amino acids 829–837) is the core sequence of a critical B-cell linear epitope of the E2 protein [62]. Building on these studies, Liu et al. further studied and expressed the fusion of the glutathione S-transferase gene (GST) and the antigen epitope in *E. coli*. The results demonstrated that the fusion protein could be used not only for specific CSFV infection diagnosis but also to induce an effective immune response. However, it could not protect all the animals infected with the C-strain challenge [63]. In this experiment, the B-cell antigenic epitopes were designed to be added to both ends of the E2 glycoprotein to form a fusion E2B, thereby enhancing adequate antigenic exposure.

Ferritin can present multiple antigenic molecules simultaneously, which typically elicit a strong immune response [27]. In this experiment, ferritin nanoparticles embedded with E2 protein and B-cell antigenic epitopes effectively induce humoral and cellular immunity. At 28 dpi, the antibody titers induced by recombinant protein E2B were slightly higher than those of the E2 alone, though the difference was not statistically significant. However, with ferritin as a carrier, the immunization effect of the ferritin-based recombinant protein E2B was comparable to that of the commercial vaccine (Figure 4). This phenomenon was also observed in the levels of the innate immune factors in vivo (Figure 5). The experimental results demonstrated that using ferritin nanoparticles as carriers to deliver recombinant proteins significantly increased the immunogenicity of antigens, inducing robust immune responses. The CSFV recombinant ferritin nanoparticles vaccine constructed for this experiment could be a candidate for preventing and controlling CSF.

## 5. Conclusions

In conclusion, this study successfully constructed fused ferritin nanoparticles with advantages in antigen presentation and structure. The dominant protein E2 of CSFV and the B-cell antigenic epitope TAVSPTTLR (amino acids 829–837) were chimerically incorporated into the vector for the first time, fully utilizing their antigenic stimulation capabilities and yielding high specific and neutralizing antibody titers. Concurrently, fused ferritin nanoparticles carrying these antigens induced heightened levels of cellular immunity. Future studies will explore combining these ferritin nanoparticles with other antigens to develop more efficient and practical vaccines.

## Figures and Tables

**Figure 1 vaccines-12-00948-f001:**
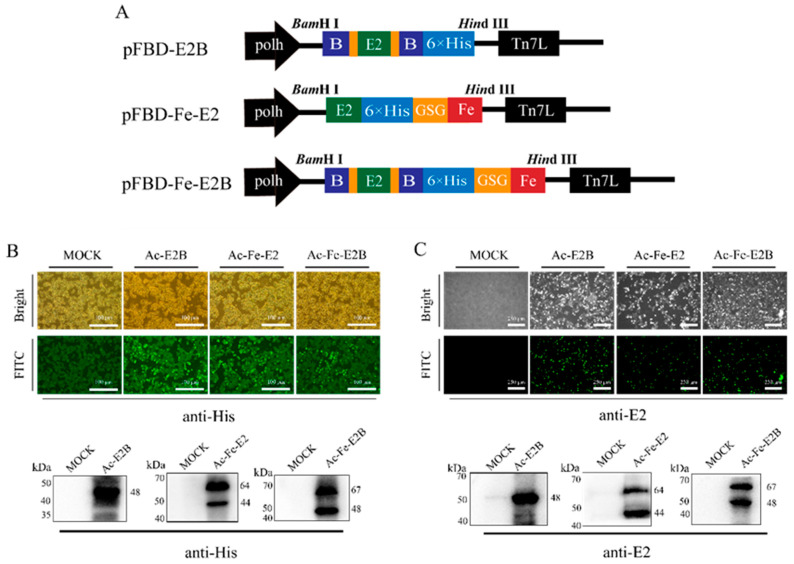
Construction of recombinant transfer plasmids and identification of recombinant proteins. (**A**) Construction of the recombinant transfer vector. In immunofluorescence assay and Western blotting experiments, the recombinant proteins were validated using anti-His antibodies (**B**) and anti-E2 antibodies (**C**).

**Figure 2 vaccines-12-00948-f002:**
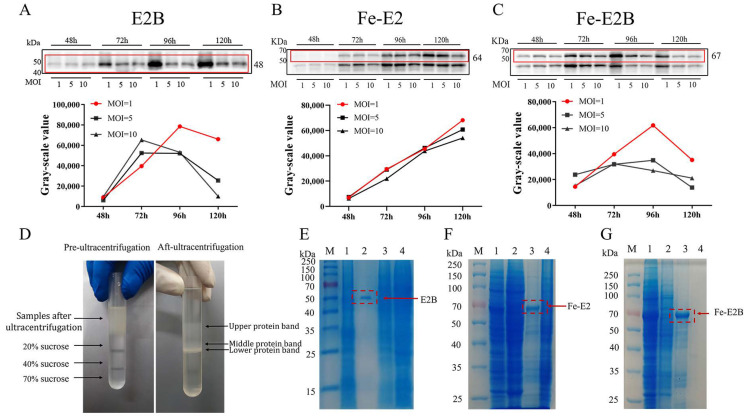
Recombinant protein expression and purification. (**A**–**C**) The protein expression trends of recombinant proteins E2B, Fe-E2, and Fe-E2B were observed under various multiplicities of infection (MOI = 1, 5, 10) of recombinant baculovirus and at different protein harvest times (48 h, 72 h, 96 h, 120 h). (**A**) Recombinant protein E2B (48 kDa). (**B**) Recombinant protein Fe-E2 (64 kDa). (**C**) Recombinant protein Fe-E2B (67 kDa). (**D**) The distribution of samples in the centrifuge tubes before and after ultracentrifugation. (**E**–**G**) Results of SDS-PAGE identification of the three protein bands after ultracentrifugation of each sample. (**E**) E2B samples. (**F**) Fe-E2 samples. (**G**) Fe-E2B samples. M: Marker. 1: Samples after ultrasonic fragmentation; 2: upper protein band after ultracentrifugation; 3: middle protein band after ultracentrifugation; 4: lower protein band after ultracentrifugation. The bands within the red boxes are the areas for grayscale value analysis.

**Figure 3 vaccines-12-00948-f003:**
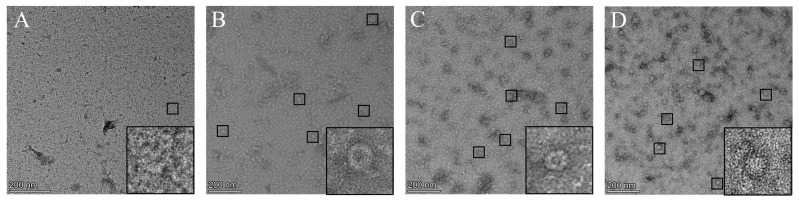
Observation of the morphology of different ferritin nanoparticles under TEM. (**A**) Morphology of proteins (E2B) not containing ferritin under TEM. (**B**) Morphology of ferritin nanoparticle alone. (**C**) Morphology of ferritin-fused E2 protein nanoparticle. (**D**) Morphology of ferritin fused with E2 protein and B-cell antigenic epitopes nanoparticle.

**Figure 4 vaccines-12-00948-f004:**
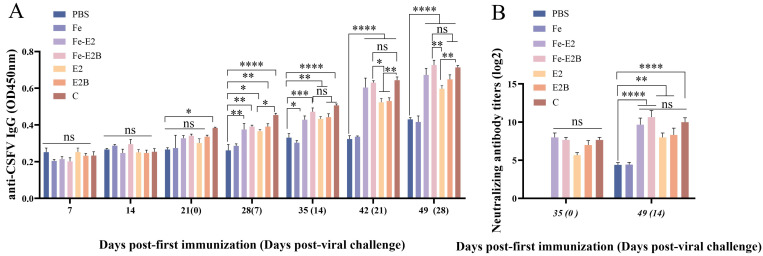
Detection of antibody titers in rabbits. (**A**) Detection of specific antibody titers in rabbit serum using ELISA. (**B**) Detection of neutralizing antibody titers in rabbit serum using the virus neutralization test on PK-15 cells. ns, *p* > 0.05; *, *p* < 0.05; **, *p* < 0.01; ***, *p* < 0.001; ****, *p* < 0.0001.

**Figure 5 vaccines-12-00948-f005:**
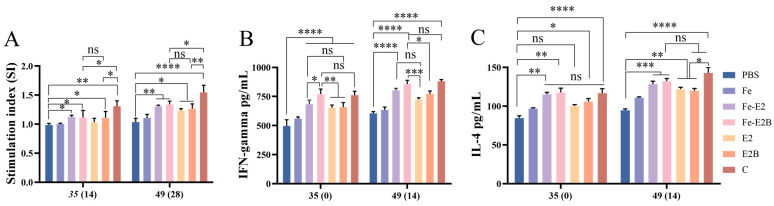
Assessment of lymphocyte proliferation and cytokine levels. (**A**) Lymphocyte proliferation assays were conducted using PBMC isolated from rabbits at 35 and 49 dpi. Sera of the experimental rabbits were collected at 35 and 49 dpi, and ELISA measured the levels of IFN-γ (**B**) and IL-4 (**C**). ns, *p* > 0.05; *, *p* < 0.05; **, *p* < 0.01; ***, *p* < 0.001; ****, *p* < 0.0001.

**Figure 6 vaccines-12-00948-f006:**
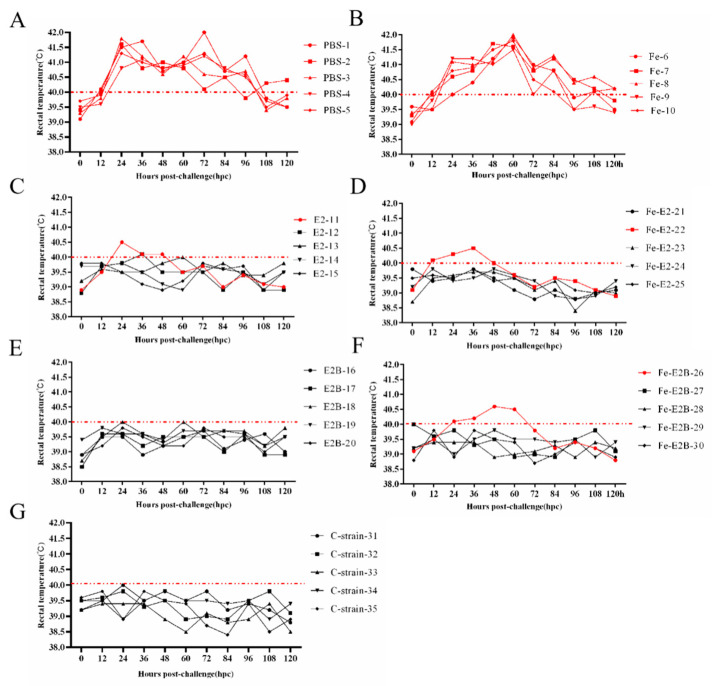
Rectal temperature change curves in rabbits challenged with the C-strain over 120 continuous hours. The body temperature changes of five rabbits in each experimental group were plotted over the 120 h. (**A**) PBS group (No. 1–5). (**B**) Fe group (No. 6–10). (**C**) E2 group (No. 11–15). (**D**) Fe-E2 group (No. 15–20). (**E**) E2B group (No. 20–25). (**F**) Fe-E2B group (No. 25–30). (**G**) C group (No. 31–35).

**Table 1 vaccines-12-00948-t001:** Animals immunization protocol.

Groups	Numbers of Rabbits	Vaccination	Type and Composition	Immune Dose	Immunization Pathway	Number of Injection	Challenge
1	5	PBS	PBS	1 mL (PBS)	IM	2 (21 dpi, Booster immunization)	100 RID_50_ C-strain (35 dpi, IV, 1dose)
2	5	Fe	W/O/W ISA 201 VG	1 mL (40 µg, purified)			
3	5	Fe-E2	W/O/W ISA 201 VG	1 mL (40 µg, purified)			
4	5	Fe-E2B	W/O/W ISA 201 VG	1 mL (40 µg, purified)			
5	5	E2	W/O/W ISA 201 VG	1 mL (40 µg, purified)			
6	5	E2B	W/O/W ISA 201 VG	1 mL (40 µg, purified)			
7	5	C-strain vaccine	C-strain attenuated vaccine (commercial vaccine)	1 dose (commercial vaccine)			

**Table 2 vaccines-12-00948-t002:** Detection of CSFV viral load in lymph nodes, spleen, and peripheral blood using qPCR in experimental rabbits after challenge.

Groups	Number	C-Strain Viral Load (copies/μL)		
		Spleen	Lymph Nodes	Peripheral Blood
PBS	PBS-1	2.37 × 10^3^	3.64 × 10^3^	0.98 × 10^3^
	PBS-3	3.45 × 10^3^	1.98 × 10^3^	2.23 × 10^3^
	PBS-5	4.32 × 10^3^	3.12 × 10^3^	2.36 × 10^3^
Fe	Fe-7	5.26 × 10^3^	3.29 × 10^3^	3.12 × 10^3^
	Fe-9	5.23 × 10^3^	2.17 × 10^3^	2.98 × 10^3^
	Fe-10	2.13 × 10^3^	4.23 × 10^3^	3.01 × 10^3^
E2	E2-12	-	-	-
	E2-14	-	-	-
	E2-15	-	-	-
E2B	E2B-17	-	-	-
	E2B-18	-	-	-
	E2B-20	-	-	-
Fe-E2	Fe-E2-22	-	-	-
	Fe-E2-24	-	-	-
	Fe-E2-25	-	-	-
Fe-E2B	Fe-E2B-26	-	-	-
	Fe-E2B-27	-	-	-
	Fe-E2B-29	-	-	-
C	C-32	-	-	-
	C-34	-	-	-
	C-35	-	-	-

Spleen, lymph nodes, and peripheral blood samples were collected at 49 dpi, and RNA was extracted using the RNA isolation kit. Viral loads were determined by qPCR as described previously. “-” indicates that it was not detected.

## Data Availability

All the data and materials can be found in the manuscript. Other requirements can be obtained by contacting the author.

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
