# Peer review of "Development of a Ferritin-Based Nanoparticle Vaccine against Classical Swine Fever"

_vaccines, 2024, doi:10.3390/vaccines12080948_

Round 1

Reviewer 1 Report

Comments and Suggestions for Authors

The study highlights the potential of ferritin nanoparticles as antigen-presenting carriers to induce robust immune responses. The Fe-E2B vaccine demonstrated enhanced immunogenicity, providing a promising strategy for CSF prevention and control.

Here are some suggestions and questions:

1.        Provide full name for some abbreviations: Sf9 cells, Hf5 cells, PK-15 cells (line 88), IM (Table 1)

2.        Line 169, mentioned the vaccine contained 40 ug of purified protein, while Table 1 states 40 uL under the Immune Dose column. The unit is not consistent.

3.        Briefly discuss why choose to use ISA 201VG adjuvant

4.        How to determine the use of 40 ug of purified protein for vaccine formulation? What’s the amount of protein for commercial C-strain attenuated vaccine? Only mentioned 1 dose, is that 1 mL as well? This could be quite important question as it will impact the immune response.

5.        A following suggestion: toxicity study is preferred as the vaccine will be injected into the organism. A single dose or repeat dose, monitor any signs of toxicity of the vaccine (for example weight), and H&E stain for major organs.

6.        When performing a comparison (Figures 4 and 5), Fe-E2B vs E2B and Fe-E2B vs C strain could be more important, suggest adding statistical analysis for the two groups in figure.

Reviewer 2 Report

Comments and Suggestions for Authors

The authors compare CSF recombinant E2 baculovirus expressed proteins with an additional neutralizing epitope and ferritin to generate VLPs. The vaccine candidates were evaluated in rabbits and a CSF challenge was done using the C strain of CSF (vaccine). 

This model and challenge are not ideal. The experiment would be much better if pigs were used and the CSF challenge was performed using a virulent CSF strain to show protection and compare the protection to the C-strain vaccine.

The results do not show any that there was any difference between the different E2 protein vaccines with respect to neutralizing antibodies following immunization and no clinical differences between them following challenge.

Line 242 Western also line 246, 249, 256 and throughout the paper.

Line 302 The ELISA kit "encapsulating" the CSFV antigen replace "encapsulating" with "containing"

Figure 4 A This is an indirect test the endpoint titres should be calculated to show differences between the antibody responses. The statistics used on the OD values are not an appropriate way to compare the differences.

Line 344 remove the term stereo-typed and throughout

Reviewer 3 Report

Comments and Suggestions for Authors

General comments

The manuscript entitled ‘Robust immune responses induced by ferritin-based nanoparticles expressing classical swine fever virus E2 protein and B-cell epitopes in rabbit model’ by Song et al. is very well-written. It clearly describes the development and validation of a new subunit vaccine based on E2 protein combined with B-cell epitopes for protection against CSFV. There were some minor remarks on the manuscript that are shared in the next paragraph ‘specific comments’.

Specific comments

L294                   BVES is there all alone on the line. Please join the abbreviation with its explanation situated on the upper part of the figure.

Figure 3              Subtitle description of Figure 3 should be arranged on the same page as the figure itself.

Figure 6              The description of Figure 6 is not together with the figure itself. Please join then on one page.

Reviewer 4 Report

Comments and Suggestions for Authors

The paper explores the development of a novel recombinant ferritin nanoparticle vaccine incorporating the CSFV E2 protein and B-cell antigenic epitopes, demonstrating enhanced immunogenicity and potential efficacy in preventing Classical Swine Fever in rabbits. The findings suggest that this vaccine could serve as a viable candidate for controlling CSF, with significant implications for veterinary medicine. It is a well-written paper that has some minor defects and formatting issues throughout the text, which should be addressed in a revised version. Some of the minor defects are:

  • Lines 2-4: The title is trying to include too much information, making it complicated and unclear. I would propose revising it to something simpler, such as "Development of a Ferritin-Based Nanoparticle Vaccine Against Classical Swine Fever".

  • Line 36: Rephrase “in several countries and regions, including Asia, Eastern Europe, …” to “in several countries, including regions in Asia, Eastern Europe, …”.

  • Line 56: Replace the abbreviation “aa” with “amino acids”.

  • Line 92: Capitalize the initial letter of the word “medium”.

  • Line 92: The location "Malmo, MA, USA" seems incorrect. The only Malmo city commonly known is in Sweden. Please verify and correct.

  • Lines 161-162: The phrase “Pentobarbital sodium analogs act as a rapid hypnotic and central depressant” is informal. Consider replacing it with “Pentobarbital sodium analogs are short-acting anesthetics”. Also, specify the exact pentobarbital sodium analogs used.

  • Line 186: "three times with PBS" - The abbreviation "PBS" (Phosphate-Buffered Saline) should be explained upon its first use in the text. Also, specify the volume and duration for each wash step.

  • Lines 337-338, 346-352: These two text sections are inserted between figures and figure legends – correct this formatting issue.

  • Line 238: “in Figure (Figure 1A)” should be “in Figure 1A”.

  • Lines 250-251: The phrase “with molecular weights of 44 kDa and 48 kDa lower than the target bands” should be “with molecular weights of 44 kDa and 48 kDa that were lower than the target bands”.

  • Line 268: Change "forming” to “formation”.

  • Line 305: "specific” should be “significant”.

  • Line 311: “between these groups” - specify the groups being referred to.

  • Line 363: "quantified using qPCR" - use the full term at its first mention (quantitative polymerase chain reaction).

  • Lines 377-378: Change "have a non-negligible disadvantage of not being able to differentiate between vaccine-immunized and wild-virus-infected animals (DIVA) serologically" to “cannot differentiate between vaccine-immunized and wild-virus-infected animals (DIVA) serologically".

  • Line 381: Change "However, it has not proven as effective in immunization as the C strain" to "However, the E2 protein has not proven as effective in immunization as the C-strain live attenuated vaccine".

  • Lines 383-385: "In this study, we designed a recombinant ferritin nanoparticle vaccine that utilizes the E2 protein and B-cell antigenic epitopes as antigens, with ferritin as a carrier, to induce a robust immune response." - While the study shows promising results, the assertion that this design will induce a "robust immune response" broadly across different conditions remains speculative without extensive field trials.

  • Line 384-385: Rephrase "with ferritin as a carrier, to induce a robust immune response" to "using ferritin as a carrier to induce a robust immune response".

  • Line 386-387: Change "high-affinity immunoglobulin G (IgG) plasma cells" to “high levels of high-affinity immunoglobulin G (IgG) produced by plasma cells".

  • Line 394: Clarify the sentence "This phenomenon may be due to feeding or environmental factors" to "This temporary increase in body temperature might be attributed to non-specific stress factors".

  • Line 400: Define the acronym BVES (baculovirus expression system).

  • Lines 403-404: Clarify "The neutralizing antibody titers in the Fe-E2 group were 1.2-fold higher than the E2 group" to "The neutralizing antibody titers in the Fe-E2 group were 1.2 times higher than those in the E2 group".

  • Line 407-409: Improve "Th1 cells release cytokines, particularly IFN-γ, to promote lymphocyte proliferation and induce cell-mediated immune responses" to "Th1 cells release cytokines, particularly IFN-γ, which promote lymphocyte proliferation and induce cell-mediated immune responses".

  • Line 431-432: Clarify "Ferritin is capable of multivalent presentation of antigenic molecules, which usually elicit a relatively effective immune response" to "Ferritin can present multiple antigenic molecules simultaneously, which typically elicit a strong immune response".
